# Evaluation of Different Cannulation Strategies for Aortic Arch Surgery Using a Cardiovascular Numerical Simulator

**DOI:** 10.3390/bioengineering10010060

**Published:** 2023-01-03

**Authors:** Beatrice De Lazzari, Massimo Capoccia, Nicholas J. Cheshire, Ulrich P. Rosendahl, Roberto Badagliacca, Claudio De Lazzari

**Affiliations:** 1Human Movement and Sport Sciences, “Foro Italico” University of Rome, 00147 Rome, Italy; 2Leeds General Infirmary, Leeds Teaching Hospitals NHS Trust, Leeds LS1 3EX, UK; 3Department of Biomedical Engineering, University of Strathclyde, Glasgow G4 0NW, UK; 4Aortic Centre, Royal Brompton Hospital, London SW3 6NP, UK; 5Department of Clinical, Internal Anesthesiology and Cardiovascular Sciences, “Sapienza” University of Rome, 00147 Rome, Italy; 6National Research Council, Institute of Clinical Physiology (IFC-CNR), 00185 Rome, Italy; 7Faculty of Medicine, Teaching University Geomedi, 0114 Tbilisi, Georgia

**Keywords:** aortic surgery, aortic arch, three-way cannulation approach, carotid artery perfusion, pressure–volume loop, lumped-parameter model, software simulation, cardiovascular modelling

## Abstract

Aortic disease has a significant impact on quality of life. The involvement of the aortic arch requires the preservation of blood supply to the brain during surgery. Deep hypothermic circulatory arrest is an established technique for this purpose, although neurological injury remains high. Additional techniques have been used to reduce risk, although controversy still remains. A three-way cannulation approach, including both carotid arteries and the femoral artery or the ascending aorta, has been used successfully for aortic arch replacement and redo procedures. We developed circuits of the circulation to simulate blood flow during this type of cannulation set up. The CARDIOSIM© cardiovascular simulation platform was used to analyse the effect on haemodynamic and energetic parameters and the benefit derived in terms of organ perfusion pressure and flow. Our simulation approach based on lumped-parameter modelling, pressure–volume analysis and modified time-varying elastance provides a theoretical background to a three-way cannulation strategy for aortic arch surgery with correlation to the observed clinical practice.

## 1. Introduction

Aortic disease has a significant impact on quality of life. Aneurysmatic dilatation and acute aortic dissection involving the aortic arch are the most devastating manifestations of aortic disease. Surgical or endovascular treatments for these conditions are currently available options. The outcome following successful total arch repair is satisfactory, but it still carries the risk of potentially devastating perioperative complications. Conventional aortic arch replacement can be offered to the majority of patients, although hybrid and endovascular techniques have gained popularity. Nevertheless, a less invasive approach can often be as technically challenging as open surgery, with stroke and endo-leaks as early limiting factors and less favourable mid- to long-term outcomes [1]. Mid-term outcomes and intra-operative complication rates with both hybrid and conventional aortic arch surgery remain heterogeneous and depend on centre experience and patient suitability [2]. Open repair with the elephant trunk technique [3] under hypothermic circulatory arrest is widely used with satisfactory long-term outcomes [4,5]. The frozen elephant trunk technique using the Thoraflex^TM^ (Vascutek, Terumo, Inchinnan, Glasgow, UK) or the E-vita OPEN PLUS (JOTEC, Hechingen, Germany) hybrid device has recently become a popular choice, which allows subsequent endovascular procedures [6,7]. Hybrid repair involves different techniques to debranch the aortic arch and create a suitable landing zone for an additional or staged endovascular procedure. Although aortic arch surgery has progressed significantly since early attempts, [8,9], cerebral protection remains a matter of debate. Due to reduced oxygen demand and cerebral metabolism, profound hypothermia (18 °C) with total circulatory arrest as the only mode of protection has been widely accepted [10], although at the expense of some neurological injury as the metabolism is never reduced to zero [11]. The use of retrograde cerebral perfusion as an additional protective measure [12] has been challenged by its unpredictable effects and ability to provide adequate cerebral capillary perfusion [13,14,15,16,17]. Although retrograde cerebral perfusion is still used by some groups [18,19], antegrade selective cerebral perfusion has gained wide acceptance and popularity [20,21,22]. The combination of antegrade selective cerebral perfusion with moderate hypothermia [20,23] has reduced the potential for neurological injury and allowed more time for repair [24,25,26]. Controversy remains whether unilateral or bilateral perfusion should be more appropriate [27,28,29] in the presence of completeness of the circle of Willis [30]. The “branch-first” technique without circulatory arrest or deep hypothermia has also been proposed [31,32].

This surgical approach consists of techniques involving different cannulation sites. When central aortic cannulation is not feasible, arterial cannulation through one of the femoral arteries (either directly or via an end-to-side Dacron graft) is appropriate. Axillary artery cannulation [33] is gaining interest, but it is not without other risks and complications [1,34]. The innominate [35] and common carotid arteries [36] have also been considered as cannulation sites. A three-way cannulation approach, which can be used for complex and redo surgery in particular, has been recently proposed [37].

We have developed circuits of the circulation to simulate blood flow during a three-way cannulation approach consisting of different configurations, including both common carotid arteries and either the femoral artery or the ascending aorta. CARDIOSIM© was the numerical simulator platform used for this purpose [38,39,40]. The aim was to analyse the effect on haemodynamic and energetic parameters and the benefit derived in terms of pressure and flow. Right atrial cannulation to ascending aorta return remains the standard approach in routine cardiac surgery, although it may be used during aortic arch surgery within certain limits. Nevertheless, an alternative cannulation configuration is necessary to preserve cerebral circulation during aortic arch replacement. The three-way cannulation approach recently proposed [37] relies on long-standing clinical experience with favourable results, which have given the motivation to develop its theoretical background and quantitatively analyse its effect on circulation. Our approach consisted of pressure–volume analysis combined with the modified time-varying elastance theory and 0-D modelling, which addresses space dependence by splitting the cardiovascular system into compartments. Physiological variables such as pressure, flow and resistance are assumed spatially uniform with variation as a function of time. Models based on the pressure–volume relationship and lumped-parameter representation of the cardiovascular system are suitable for a clinical setting, given their simplicity and versatility [41,42,43,44,45]. Although they provide less detailed predictions of pressure and flow waveforms, these models have shown great flexibility in simulating the haemodynamics of different cardiovascular conditions and therapeutic interventions with the potential to be run in real time on desktop, laptop or mobile devices [46]. The solution time is practically immediate, which is once again quite suitable for a clinical setting. The 3-element Windkessel model is quite often used to represent the compliant and resistive nature of arterial circulation [47]. Additionally, it is quite frequently used to improve boundary conditions for 3D models of arterial, ventricular or venous pathophysiology [47,48,49,50]. Patient-specific modelling is currently driving the development of more personalised treatment approaches to guide medical and surgical intervention in terms of optimisation and outcome prediction [51,52,53,54]. A recent review has highlighted the critical role played by lumped-parameter models in patient-specific cardiovascular modelling [55].

## 2. Materials and Methods

### 2.1. Cannulation Strategies for Aortic Arch Surgery

Figure 1 shows four different cannulation strategies for aortic arch surgery as the main subject of this study. The centrifugal pump draws blood and ejects it according to the following connections [56,57,58]:→

Right atrial (RA) cannulation to ascending aorta (AA) and bilateral common carotid artery (CC) return (top-left panel), RA→AA&CC.Right atrial cannulation to femoral artery (FA) and bilateral common carotid artery return (top-right panel), RA→FA&CC.Femoral vein (FV) and superior vena cava (SVC) cannulation to femoral artery and bilateral common carotid artery return (bottom-left panel), FV&SVC→FA&CC.Femoral vein cannulation to femoral artery and bilateral common carotid artery return (bottom-right panel), FV→FA&CC.

### 2.2. The Heart and Circulatory Numerical Network

“In silico” study of cannulation for aortic arch surgery was performed developing new 0-D numerical modules of the cardiovascular network implemented in CARDIOSIM© platform [38,59,60,61,62,63,64]. These modules allow the reproduction of the behaviour of native left and right ventricles; atria and septum; coronary circulation; ascending aorta and aortic arch; descending thoracic and abdominal aorta; upper limbs and head section; superior, inferior and abdominal vena cava section; renal and hepatic section; splanchnic and lower limbs sections; pulmonary circulation. Figure 2a shows an overview of the whole cardiovascular network. The native left and right ventricles, atria and septum and the ventricular, atrial and septal activity synchronized with the electrocardiographic (ECG) signal are implemented using the time-varying elastance concept [60,61,63,64]. This numerical representation allows the simulation of inter-ventricular and intra-ventricular dyssynchrony. The specific modules of coronary circulation are available in CARDIOSIM© platform [38]. The module of coronary circulation presented in [65] was selected for this study. Figure 2a shows two different electric analogues used to reproduce the behaviour of the aortic (AV), mitral (MV), tricuspid (TV) and pulmonary (PV) valves [38]. A representation with an ideal diode was chosen for this study: when the pressure across the valve was positive, the valve opened and allowed blood flow; when the pressure was less than or equal to zero, the valve closed, and blood flow was zero. The whole pulmonary circulation was modelled as described in the current literature [63,64,66,67].

Figure 2b shows the electric analogue of the following compartments: ascending and descending aorta, aortic arch and abdominal tract. All these sections were modelled with resistance, inertance and compliance (RLC) elements. The thoracic compartment was modelled with two resistances (R_THOR_ and R_ATI_), inertance (L_ATI_) and compliance (C_ATI_). The abdominal section was divided in two compartments both electrically represented with RLC elements. Finally, the behaviour of the superior vena cava section was reproduced with RC elements. The symbols used in Figure 2b are listed in Table 1.

The circuit representation of the upper and lower limbs and the head sections with the four different cannulation strategies implemented in CARDIOSIM© are illustrated in Figure 2c. The nomenclature of the symbols is listed in Table 1. The lower limb section consists of two parts reproducing left and right limb circulation. The left (right) arterial circulation of the lower limb is modelled with a variable resistance R_LFA_ (R_RFA_), the resistance R_LFV_ (R_RFV_) reproduces the venous circulation, and the capacitor C_LLE_ (C_RLE_) represents the left (right) compliance. The upper limbs are modelled with two resistances R_ARM1_ (variable resistance) and R_ARM2_ reproducing arterial and venous circulation, respectively, and compliance C_ARM_. The arterial and venous cerebral circulation is reproduced with a variable resistance R_HD1_ and a resistance R_HD_, respectively; the capacitor C_HD_ mimics the vessel compliance.

Figure 2d shows the electrical analogue of the second abdominal tract modelled with RLC elements. The variable resistance R_SP1_ and the resistance R_SP2_ allow the simulation of the arterial and venous circulation of the splanchnic tract; the capacitor C_SP_ mimics its compliance.

The behaviour of the arterial and venous renal (hepatic) section is simulated with a variable resistance R_KID1_ (R_HEP1_) and the resistance R_KID2_ (R_HEP2_), respectively. The compliance of the renal and hepatic vessels is modelled with the capacitors C_KID_ and C_HEP_, respectively.

### 2.3. Numerical Models of the Cannulae and Centrifugal Pump

Figure 2c shows the four different cannulation strategies implemented in CARDIOSIM© platform. In the first one, the centrifugal pump draws blood from RA and ejects it in the ascending aorta and in the common carotid artery bilaterally (RA→AA&CC). In this case, the switches SW5 (which opens the input cannula connected to the right atrium), SW3 (which opens the output cannula connected to the common carotid artery) and SW6 (which opens the output cannula connected to the ascending aorta) are ON, and the other switches are OFF. When the second type of cannulation is activated, the centrifugal pump draws blood from the right atrium and ejects it into the FA and in the common carotid artery bilaterally (RA→FA&CC). This connection is obtained by switching SW5, SW3 and SW4 (which opens the output cannula connected to the femoral artery) ON, and setting SW1, SW2 and SW6 OFF. In the connection (FV→FA&CC), the pump draws blood from the femoral vein and ejects it into the FA and CC. The cannulation can be configured by setting SW1 (which opens the input cannula connected to the femoral vein), SW3 and SW4 ON. Finally, in the last connection, the centrifugal pump draws blood from the FV and the superior vena cava and ejects it into the FA and CC. The cannulation (FV&SVC→FA&CC) can be activated by switching SW1, SW2 (which opens the input cannula connected to the SVC), SW3 and SW4 ON, whilst SW5 and SW6 are switched OFF. All the cannulae are modelled with RLC elements [63].

The numerical model of the centrifugal pump implemented in CARDIOSIM© has been previously described in [61,68].

### 2.4. Simulation Protocol

The results of the simulations performed in this work were obtained starting from baseline conditions and setting heart rate (HR), left and right ventricular and septal elastance, arterial and venous resistances and compliances for all compartments to achieve minimal physiological conditions characterized by HR = 90 beat/min, systolic aortic pressure (AoP_sys_) = 90 mmHg, LAP = 21 mmHg, RAP = 10 mmHg, PAP = 18 mmHg, CO = 4.3 l/min. Starting from baseline conditions, each cannulation configuration was activated by setting the centrifugal pump speed to 2000, 2500, 3000, 3500, 4000 and 4500 rpm. The effects induced by different cannulations on the left and right ventricular loop and on the left atrial loop are presented and discussed. Furthermore, the instantaneous waveform of the aortic and pulmonary arterial pressures for each cannulation configuration is included in the Results section. Finally, an analysis of the percentage variations in the haemodynamic and energetic variables calculated with respect to baseline conditions for each cannulation configuration and different speeds of the centrifugal pump was carried out in this study. Haemodynamic variables, such as mean aortic pressure, PAP, LPA, RAP, CO, left (right) ventricular–arterial coupling Ea_S_/Ees_Left_ (Ees_Right_/Ea_P_), cerebral, renal and superior vena cava flow, together with energetic variables, such as left and right ventricular external work (EW) and right ventricular pressure–volume area (PVA), were analysed.

## 3. Results

Figure 3 shows the effects induced on the left (top-left panel) and right (bottom-left panel) pressure–volume loops by different rotational speeds of the centrifugal pump applied when the cannulae are connected in the RA→FA&CC mode. The right panel of the figure shows the right atrial to femoral artery and bilateral common carotid artery cannulation. The different left and right ventricular loops were obtained in baseline conditions (black dashed line) and when the rotational pump speed was set to 2000, 2500, 3000, 3500, 4000 and 4500 rpm. When the rotational pump speed increased, the left (right) ventricular loop shifted to the left (right) with a decrease (increase) in left (right) ventricular end-diastolic (EDV) and end-systolic (ESV) volume.

Figure 4 shows the comparison of the four different methods of cannulation obtained setting the rotational pump speed to 2000 rpm. Left ventricular pressure–volume loops obtained using the software simulator were stored in excel files and subsequently plotted. The top panel shows the left ventricular pressure–volume loop in baseline conditions (dashed black line) following RA→AA&CC cannulation, obtained connecting the input cannula to the right atrium and the output cannulae to the ascending aorta and common carotid artery bilaterally (red line); RA→FA&CC cannulation (green line); FV→FA&CC cannulation, obtained connecting the input cannula to the femoral vein and the output cannulae to the femoral artery and common carotid artery bilaterally (blue line); and FV&SVC→FA&CC cannulation, obtained connecting the input cannulae to the femoral vein and the superior vena cava and the output cannulae to the femoral artery and common carotid artery bilaterally (lilac line). The middle (bottom) panel shows the left atrial (right ventricular) pressure–volume loops.

Figure 5 shows the aortic pressure (AoP) and pulmonary arterial pressure (PAP) waveforms simulated in baseline conditions and following FV&SVC→FA&CC cannulation with the pump rotational speed set to 2000, 3500 and 4500 rpm.

When the pump speed was set to 2000 rpm, the AoP and PAP waveforms (blue) moved downward compared to baseline conditions.

Figure 6 shows the effects induced on the right ventricular pressure–volume loop (top-left panel) and instantaneous PAP waveform (bottom-left panel) by FV→FA&CC cannulation when the rotational pump speed was set to 2000, 2500, 3000 and 3500 rpm. This type of cannulation generated limited effects compared to baseline conditions (black dashed line) when the centrifugal pump speed was set to 2000 rpm (red line). When the pump rotational speed increased, the right ventricular pressure–volume loop shifted to the right increasing both ESV and EDV. Instantaneous PAP waveforms were also affected by high pump rotational speeds.

The outcome of RA→AA&CC cannulation on the left atrial pressure–volume loop and on the aortic pressure (AoP) is shown in Figure 7. This type of cannulation produced negligible effects when the speed of the pump was set to 2000 rpm (red lines in the top- and bottom-left panels). At higher pump speeds, the loops (top left panel) move to the right for high values of the left atrial end-diastolic volume. AoP increased considerably when the pump rotational speed was set to 3000 rpm (lilac waveforms in the bottom panel).

Figure 8 shows the initial analysis of percentage variation in AoP, CO and left (right) atrial pressure LAP (RAP) for the four different cannulation strategies compared to baseline conditions when the rotational pump speed was set to 2500 rpm. The percentage variation in mean AoP, CO and LAP decreased for RA→FA&CC and FV&SVC→FA&CC cannulations. A decrease in percentage variation in the right atrial pressure was observed for FV&SVC→FA&CC cannulation (bottom-right panel).

Figure 9 shows the relative changes in cardiac output calculated in comparison to baseline conditions for the four methods of cannulation. The simulations were performed setting the pump rotational speed to 2000, 2500, 3000, 3500, 4000 and 4500 rpm. In the case of RA→AA&CC (FV→FA&CC) cannulation, CO increased up to 58% (52%) when the pump speed was set to 4500 rpm (top left panel and bottom right panel). On the contrary, in the case of RA→FA&CC cannulation, CO decreased by 17% when the pump speed was set to 4500 rpm (upper right panel).

For FV&SVC→FA&CC cannulation, the outcome of the simulations showed that cardiac output took a positive turn when the rotational pump speed was set to 3000 rpm (lower left panel). A percentage increase in excess of 32% (compared to the baseline conditions) in CO was observed when the pump speed was set to 4500 rpm.

The analysis of the two parameters that measure the coupling between the circulatory arterial network and the two ventricles and of the left and right ventricular external work is summarized in Figure 10 for all the cannulations with the pump rotational speed set to 3500 rpm.

An increase in Ea_S_/Ees_Left_ (top-left panel) for both RA→FA&CC and FV&SVC→FA&CC cannulations was observed when the pump rotational speed was set to 3500 rpm. On the contrary, the right ventricular–arterial coupling showed a percentage reduction compared to the baseline values for all types of cannulations (top-right panel). Only the RA→FA&CC cannulation configuration underwent a reduction in left ventricular EW of less than 2%. An EW increase of up to 7% was observed for the RA→AA&CC cannulation method (bottom-left panel). When the pump speed was set to 3500 rpm, a modest percentage increase in right ventricular EW compared to baseline conditions (bottom right panel) was observed for all the cannulation configurations.

An analysis of the percentage changes in left ventricular EW (right ventricular PVA) for FV→FA&CC, FV&SVC→FA&CC and RA→AA&CC cannulation configurations when the pump rotational speed was set to 2000, 2500, 3000, 3500, 4000 and 4500 rpm is shown in the top-left (right) panel of Figure 11. When the pump speed was set to 4500 rpm, an increase of more than 15% in left ventricular EW and about 25% in right ventricular PVA was observed for the RA→AA&CC cannulation configuration.

Percentage variation in RAP (LAP) compared to basal.

Baseline conditions are available in the bottom-left (right) panel for FV→FA&CC, FV&SVC→FA&CC and RA→AA&CC cannulation configurations simulated by setting the pump rotational speed to 2000, 2500, 3000 and 3500 rpm. A percentage reduction in LAP (bottom-right panel) was observed for RA→AA&CC cannulation with each pump rotational speed.

Figure 12 shows the effects induced by the four different cannulation strategies on cerebral blood flow when the pump rotational speed was set to 3500 rpm (top left panel). The top-right panel shows the effect induced by RA→FA&CC cannulation on cerebral blood flow, superior vena cava flow and CO when the pump speed was set to 2500 rpm. The relative changes in cerebral blood flow*, SVC* and CO* were evaluated when a little increase in the resistance of the ascending aorta tract was simulated.

The effects induced by the RA→FA&CC cannulation strategy on PAP and LAP (kidney and hepatic flow) with the pump rotational speed set to 3500 (2500) rpm are shown in the bottom-left (right) panel. The percentage variation in kidney flow*, hepatic flow* (bottom-right panel), PAP* and LAP* (bottom-left panel) were evaluated when a little increase in ascending aorta resistance was considered during the simulation.

The effects induced by different cannulation strategies on SVC and hepatic flow were further analysed. Figure 13 shows the relative changes in SVC (top-left panel) and hepatic flow (bottom-left panel) obtained when the rotational pump speed was set to 3500 rpm. A percentage decrease was observed during the RA→FA&CC cannulation configuration. The top (bottom)-right panel shows the relative changes in cerebral (kidney, hepatic and SVC) blood flow obtained when the FV&SVC→FA&CC cannulation method was simulated with different rotational pump speeds. A trend reversal was observed when the pump speed is greater than or equal to 3000 rpm.

## 4. Discussion

The management of aortic arch disease either in the context of acute type A aortic dissection or in the presence of aneurysmatic disease is quite challenging and still a matter of significant debate [69,70]. Open repair for aortic arch disease remains the standard of care in high volume centres, although endovascular treatment has become quite an established approach in the presence of significant comorbidities [71]. The protection of cerebral blood flow remains the key issue during aortic arch replacement. Experimental evidence confirms the benefit of selective antegrade cerebral perfusion compared to retrograde perfusion and hypothermic circulatory arrest [72]. The effect of arterial filtration on cerebral auto-regulation [73,74] and the choice of pH management [75] has also been addressed. A combined experimental and computational approach may have a role to play with the optimisation of cerebral blood flow on cardiopulmonary bypass in terms of a reduction in neurological events [76,77]. The presence of carotid or intracranial atherosclerotic disease does not seem to affect the incidence of stroke during aortic arch surgery, whilst its perioperative cause remains predominantly embolic [78]. Impaired autoregulation of cerebral blood flow during CPB may predispose to hypoperfusion and increase the risk of perioperative stroke following cardiac surgery [79,80], although a threshold remains difficult to determine [81]. Continuous monitoring with near-infrared spectroscopy (NIRS) is an established approach [82,83], with particular application during aortic arch surgery. The aortic arch is relatively free of atherosclerosis due to the helical flow pattern induced by aortic torsion, which stabilizes blood flow in the aorta and compensates the adverse effects related to aortic curvature [84]. The computational fluid dynamics modelling of blood flow has given insights into the physiological flow interactions in the health and disease of the circulatory system [85,86]. Adequate perfusion remains critical for organ preservation during cardiopulmonary bypass with particular reference to cerebral circulation. Our approach aimed to give a theoretical and quantitative background to an established technique in clinical practice [37]. Although right atrial cannulation to aortic return remains the standard and most appropriate approach in routine cardiac surgery, it may not be completely applicable to aortic surgery on a regular basis. Therefore, an alternative configuration is necessary to address the needs of aortic arch surgery. Our findings suggest that RA→AA&CC and FV→FA&CC cannulation configurations increase and maintain cerebral perfusion appropriately up to 27% and 14% (Figure 12), confirming their adaptability according to the clinical setting. RA→AA&CC cannulation would be quite appropriate for an elective aortic arch replacement due to severe aneurysmatic dilatation, whilst FV→FA&CC would be more appropriate in the presence of significant haematoma and dissection involving the whole aortic arch. An additional arterial cannula can be included during the setting for cardiopulmonary bypass and used later to restore antegrade flow through the side branch of the Dacron graft after completion of distal anastomosis during the reconstruction of the aortic arch. RA→FA&CC and FV&SVC→FA&CC are suitable alternatives but more dependent on pump setting and flow conditions (Figure 9 and Figure 13). RA→AA&CC and FV→FA&CC cannulation configurations increase cardiac output up to 58% and 52% (Figure 8 and Figure 9). The two cannulation configurations are consistent with increased cardiac output following a stepwise increase in pump rotational speed. The PV-loop analysis shows the appropriate unloading of the left ventricle whilst the right ventricle seems less affected (Figure 4). This would be addressed by suction, as required in routine cardiac surgery. FV&SVC→FA&CC is consistent with increased cardiac output when the pump rotational speed is at least 3000 rpm or above. A progressive decrease in cardiac output was observed with stepwise increase in pump rotational speed following the RA→FA&CC cannulation configuration. Despite its criticism, 0-D (lumped-parameter) models rely on the geometrical and mechanical properties of the constitutive elements and require low computational cost. The advantages of a 0-D model consist of a global representation of cardiovascular dynamics in the whole circulatory system and the evaluation of pressure and flow changes in a local circulation loop. The disadvantages are related to an inability to simulate the effect of high frequency components of arterial impedance and accurately match the aortic pressure and flow waveforms. Pressure and flow changes in specific segments and venous pressure fluctuations cannot be described; the pulse wave transmission effect cannot be simulated. Nevertheless, 0-D models remain an excellent tool that provides a simplified and effective representation of the cardiovascular system and strongly contributes to our understanding of cardiovascular physiology with great potential for clinical application [87,88,89,90]. A key aspect is the achievement of a compromise between the level of complexity and the pragmatic limitation of data collection in a clinical setting to allow appropriate personalised management [91]. In addition, a non-invasive and patient-specific approach may be beneficial for the estimation of haemodynamic variables that cannot be evaluated directly from clinical measurements [92,93]. Needless to say, 0-D,-1-D and 0-D-3-D coupling remains an important modelling application [94,95]. Although we acknowledge the limitations of our work, initial results confirm the clinical observation. The quantitative evaluation is quite reliable considering that our software has been previously validated in terms of accuracy and predictive performance [96]. Bilateral carotid artery cannulation through Dacron tubular grafts anastomosed in an end-to-side fashion allows continuous cerebral perfusion with great flexibility and regardless of the completion of the circle of Willis [37]. Stroke remains the most feared complication whether an open, hybrid or total endovascular approach is used, and its cause is multi-factorial. The increased use of selective antegrade cerebral perfusion (SACP) with warmer circulatory arrest (24–28 °C) has contributed significantly to improved stroke and lower death rates following open surgery, although there is no clear consensus about the optimal protection strategy. The three-way cannulation approach here described is a sophisticated approach, which may have a role to play for complex aortic arch and redo procedures requiring safe and prolonged antegrade cerebral perfusion to allow more surgical time. Needless to say, a close co-operation between cardiac and vascular surgeons and interventional radiologists remains essential to address aspects that are not completely within the domain of individual specialists.

## 5. Conclusions

The outcome of our simulations suggests adequate haemodynamics and organ perfusion pressure when the three-way cannulation approach is used in RA→AA&CC and FV→FA&CC cannulation configurations. This is consistent with the observed clinical practice, which may be considered an indirect validation of our simulation approach. The other two cannulation configurations, RA→FA&CC and FV&SVC→FA&CC, may be considered as alternative option, although less attractive, due to their more pronounced dependence on pump settings and reduced beneficial effect on cerebral circulation. Nevertheless, further work is required to address other aspects of the cannulation set up here described, which may help improve its clinical application.

## Figures and Tables

**Figure 1 bioengineering-10-00060-f001:**
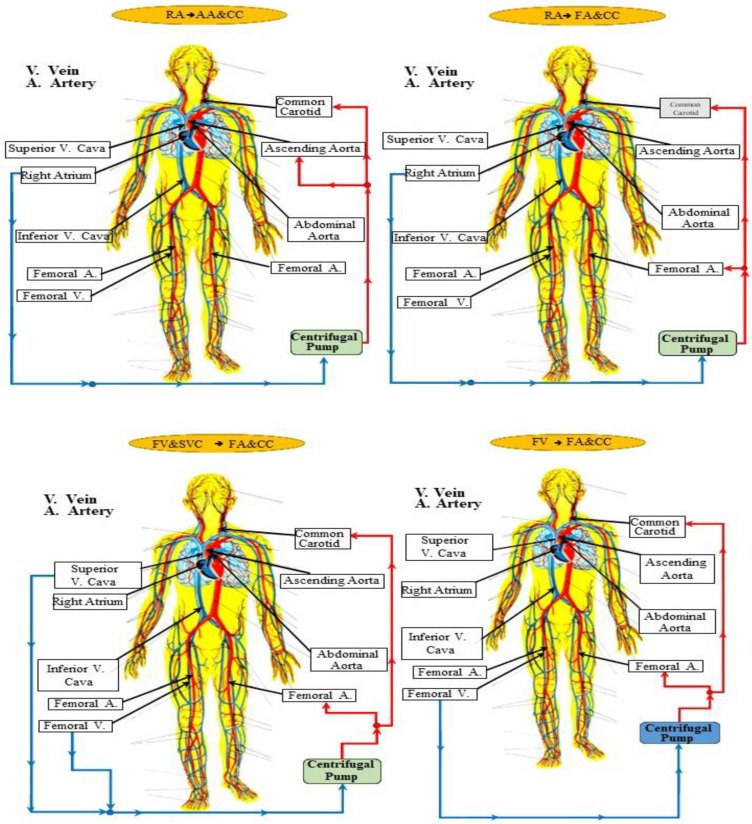
Schematic representation of the four cannulation strategies: right atrial to ascending aorta and bilateral common carotid artery return (**top left**); right atrial to femoral artery and bilateral common carotid artery return (**top right**); femoral vein and SVC to femoral artery and bilateral common carotid artery return (**bottom left**); and femoral vein to femoral artery and bilateral common carotid artery return (**bottom right**).

**Figure 2 bioengineering-10-00060-f002:**
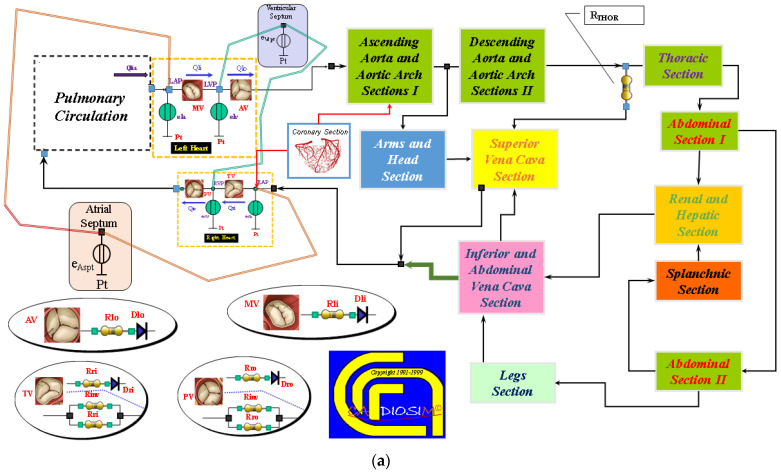
(**a**) Schematic representation of the cardiovascular system implemented in CARDIOSIM© software platform. (**b**) Electric analogue of the ascending and descending aorta, aortic arch, thoracic and abdominal sections and superior vena cava. (**c**) Electric analogue of the upper and lower limbs and the head. The six switches allow the activation of one of the four cannulation strategies. (**d**) Electric analogue of abdominal section (II), splanchnic, renal, hepatic, inferior and abdominal vena cava sections.

**Figure 3 bioengineering-10-00060-f003:**
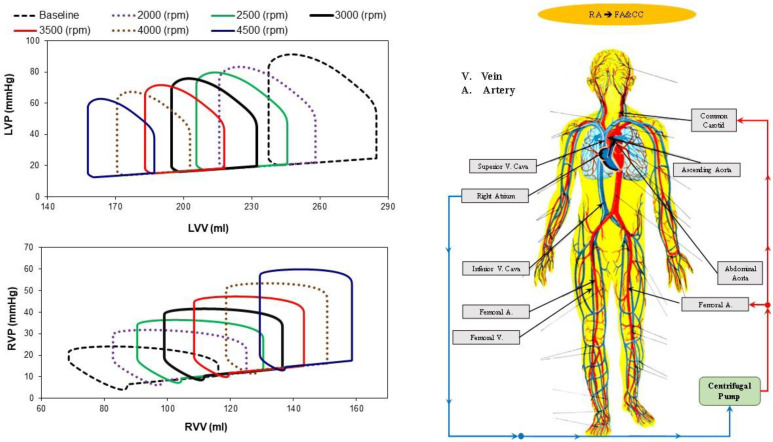
**Top left panel** (**bottom left panel**) shows the left (right) ventricular pressure–volume loops in baseline conditions (dashed black line) and for RA→FA&CC cannulation with centrifugal speed pump set to 2000 rpm (dashed lilac line), 2500 rpm (green line), 3000 rpm (continuous black line), 3500 rpm (red line), 4000 rpm (dashed brown line) and 4500 rpm (lilac line). The ventricular loops obtained using CARDIOSIM© were stored in excel files and subsequently plotted. The (**right panel**) shows the RA→FA&CC cannulation in which the centrifugal pump draws blood from the right ventricle and ejects it in the femoral artery and common carotid artery bilaterally.

**Figure 4 bioengineering-10-00060-f004:**
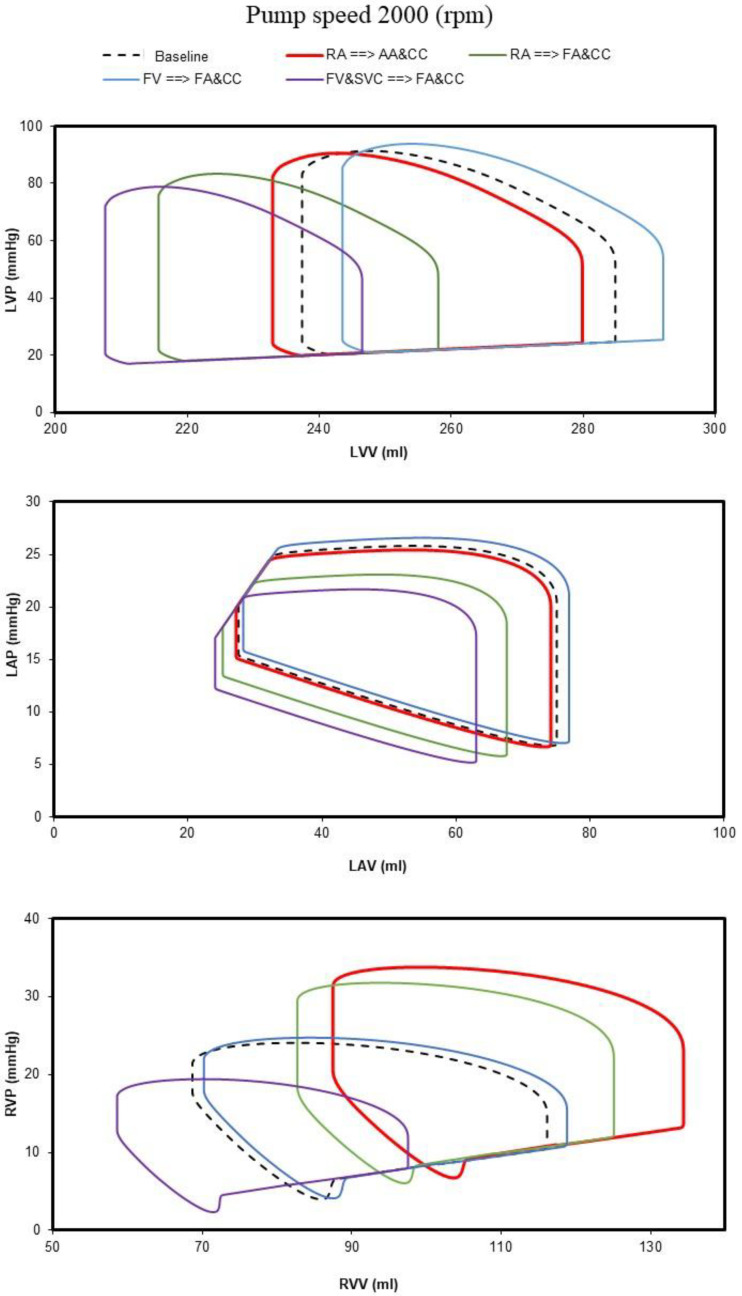
The **top** (**middle**) **panel** shows the left ventricular (atrial) pressure–volume loop obtained in baseline conditions and following the four different methods of cannulation when the centrifugal pump speed was set to 2000 rpm. The (**bottom panel**) shows the right ventricular pressure–volume loops. The dashed black loop is obtained in baseline conditions. The red, green, light blue and lilac loops are obtained following RA→AA&CC, RA→FA&CC, FV→FA&CC and FV&SVC→FA&CC connection, respectively.

**Figure 5 bioengineering-10-00060-f005:**
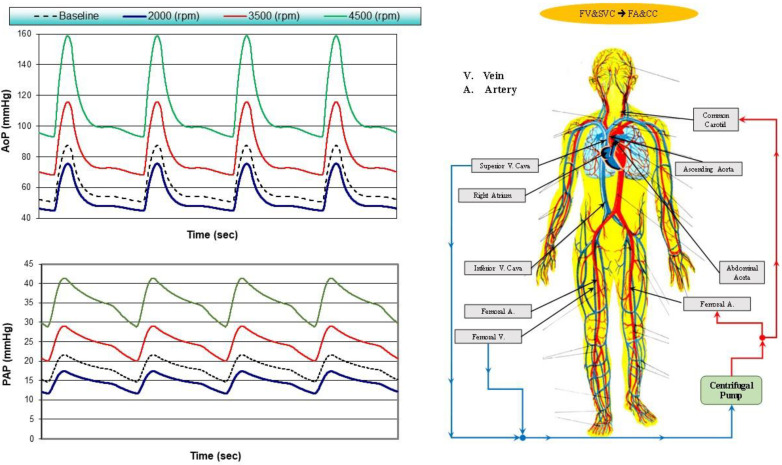
AoP (**top-left panel**) and PAP (**bottom-left panel**) waveforms obtained in baseline conditions and with FV&SVC→FA&CC cannulation when the pump rotational speed was set to 2000, 3500 and 4500 rpm. The (**right panel**) shows the FV&SVC→FA&CC cannulation in which the centrifugal pump has two input cannulae (from superior vena cava and femoral vein) and two output cannulae (to femoral and common carotid arteries).

**Figure 6 bioengineering-10-00060-f006:**
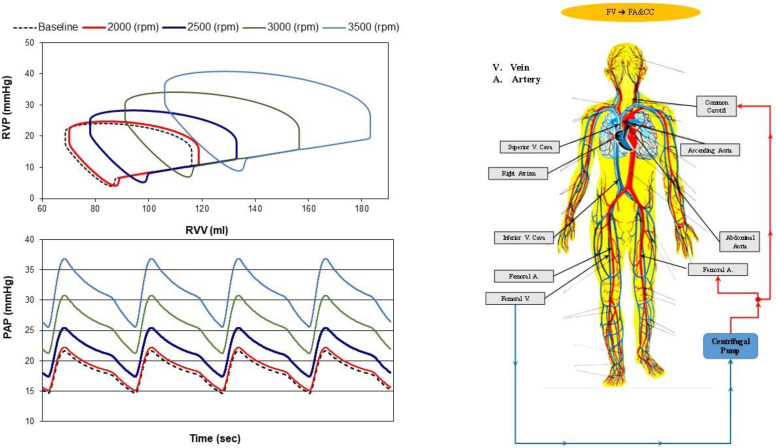
Right ventricular pressure–volume loops (**top left panel**) and instantaneous PAP waveforms (**bottom left panel**) obtained in baseline conditions (black dashed line) and with FV→FA&CC cannulation when the pump rotational speed was set to 2000 (red line), 2500 (blue line), 3000 (green line) and 3500 (light blue line) rpm. The ventricular loops and the instantaneous waveforms obtained using CARDIOSIM© were stored in excel files and subsequently plotted. In FV→FA&CC cannulation (**right panel**), the centrifugal pump draws blood from the femoral vein and ejects it into the femoral artery and common carotid artery bilaterally.

**Figure 7 bioengineering-10-00060-f007:**
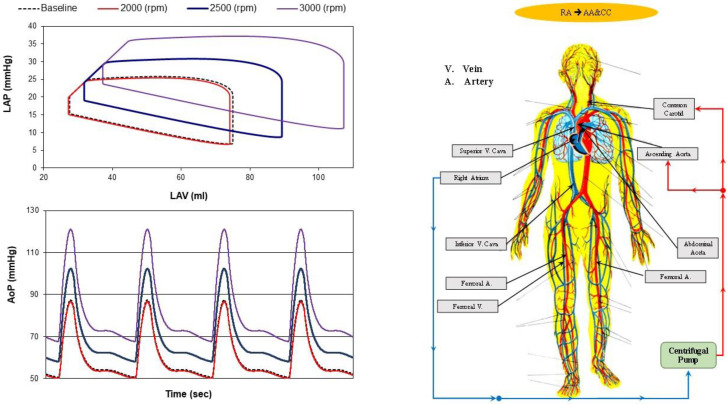
Left atrial pressure–volume loops (**top-left panel**) and instantaneous AoP waveforms (**bottom-left panel**) obtained in baseline conditions (black dashed line) and with RA→AA&CC cannulation when the pump rotational speed was set to 2000 (red line), 2500 (blue line) and 3000 (lilac line) rpm. The ventricular loops and the instantaneous waveforms obtained using CARDIOSIM© were stored in excel files and subsequently plotted. In RA→AA&CC cannulation (**right panel**) the centrifugal pump draws blood from the right atrium and ejects it into the ascending aorta and common carotid artery bilaterally.

**Figure 8 bioengineering-10-00060-f008:**
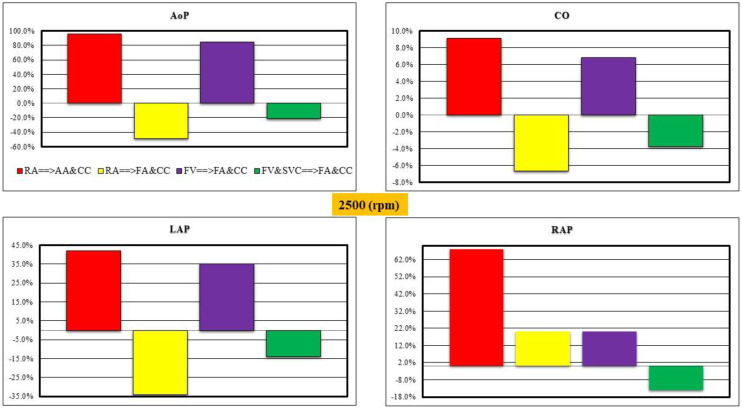
Relative changes calculated in comparison to baseline conditions for the four different methods of cannulation with pump rotational speed set to 2500 rpm. The **top** (**bottom**)**-left panel** shows the relative changes in the aortic pressure AoP (left atrial pressure). The **top** (**bottom**)**-right panel** shows the relative changes in the cardiac output CO (right atrial pressure).

**Figure 9 bioengineering-10-00060-f009:**
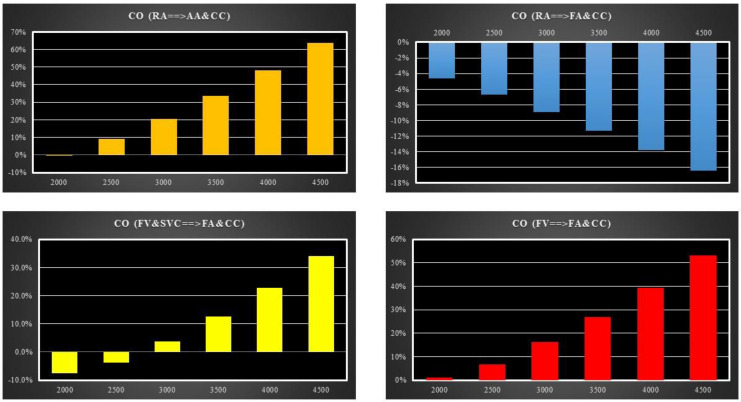
Relative CO changes calculated in comparison to baseline conditions for the four different cannulation strategies at different pump rotational speed. The **top** (**bottom**)**-left panel** shows the relative changes in CO induced by RA→AA&CC (FV&SVC→FA&CC) cannulation. The **top** (**bottom**)**-right panel** shows the relative changes in cardiac output induced by RA→FA&CC (FV→FA&CC) cannulation.

**Figure 10 bioengineering-10-00060-f010:**
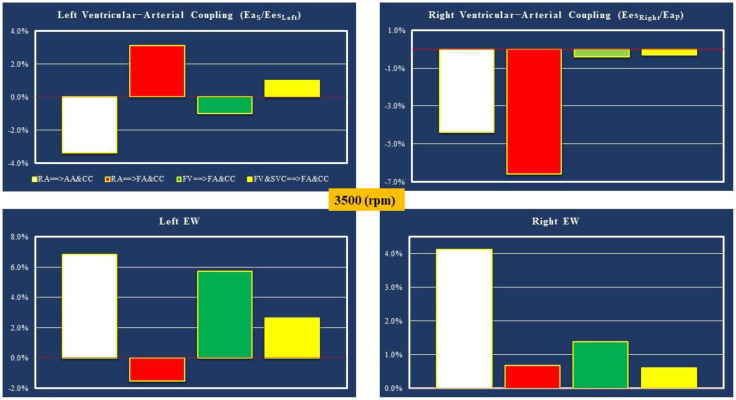
The **top-left** (**right**) **panel** shows the relative changes in left (right) ventricular–arterial coupling Ea_S_/Ees_Left_ (Ea_P_/Ees_Right_) for different cannulations applied with pump rotational speed set to 3500 rpm. The relative changes estimate with respect to baseline conditions of the energetic variable (EW) for the left (right) ventricle is reported in the **bottom-left** (**right**) **panel**.

**Figure 11 bioengineering-10-00060-f011:**
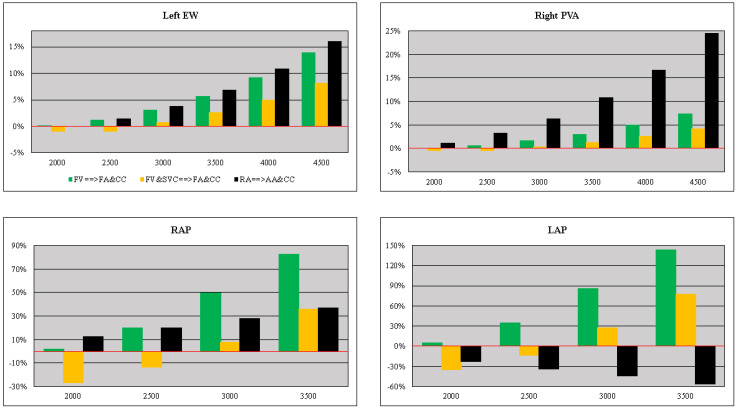
**The top-left** (**right**) **panel** shows the relative changes in left (right) ventricular EW (PVA) observed for three different cannulations with pump rotational speed set to 2000, 2500, 3000, 3500, 4000 and 4500 rpm. **The bottom left** (**right**) **panel** shows the relative changes in right (left) atrial pressure compared to baseline conditions for FV→FA&CC, FV&SVC→FA&CC and RA→AA&CC cannulation configurations with pump rotational speed set to 2000, 2500, 3000 and 3500 rpm.

**Figure 12 bioengineering-10-00060-f012:**
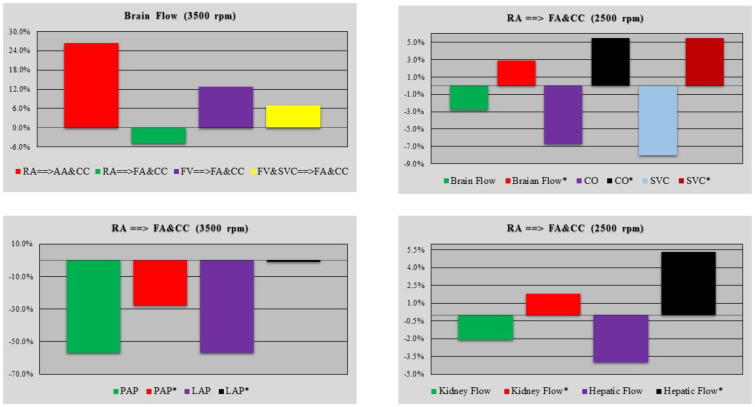
The (**top-left panel**) shows the relative changes in cerebral blood flow induced by three different cannulation configurations with pump rotational speed set to 3500 rpm. The relative changes in cerebral blood flow, superior vena cava flow and cardiac output for RA→FA&CC cannulation with pump speed set to 2500 rpm are available in the (**top-right panel**). Relative changes in CO*, cerebral blood flow* and SVC* blood flow were evaluated when a little increase in ascending aorta resistance was considered during the simulation. Kidney flow*, hepatic flow* (**bottom-right panel**), PAP* and LAP* (**bottom-left panel**) were evaluated in the same conditions when the pump rotational speed was set to 2500 and 3500 rpm, respectively.

**Figure 13 bioengineering-10-00060-f013:**
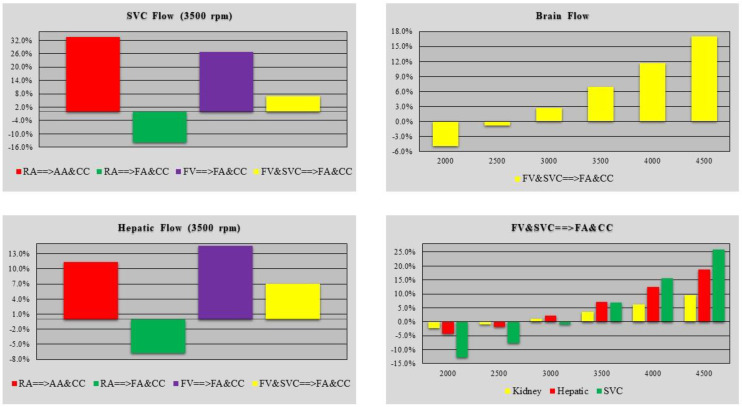
The top (bottom)-left panel shows the relative changes in SVC flow (hepatic flow) induced by four different cannulation strategies obtained when the pump rotational speed was set to 3500 rpm. The top (bottom)-right panel shows the cerebral flow (kidney, hepatic and SVC flow) for FV&SVC→FA&CC cannulation method when the pump rotational speed was set to 2000, 2500, 3000, 3500, 4000 and 4500 rpm.

**Table 1 bioengineering-10-00060-t001:** Symbols.

Symbol	Description	Unit
AoP (AAP)	Aortic (ascending and aortic) pressure	mmHg
LAP (RAP)	Left (right) atrial pressure	mmHg
LVP (RVP)	Left (right) ventricular pressure	mmHg
DAP (SVCP)	Descending aortic (superior vena cava) pressure	mmHg
THP (ABP_I_)	Thoracic (abdominal) pressure	mmHg
HDP (ARP)	Brain (Arm) pressure	mmHg
LLEP (RLEP)	Left (right) leg pressure	mmHg
Pt	Intrathoracic pressure	mmHg
P_B_	Breathing pressure	mmHg
SP (ABP_II_)	Splanchnic (abdominal II) pressure	mmHg
HP (KP)	Hepatic (renal) pressure	mmHg
IVCP	Inferior vena cava pressure	mmHg
R_AA1_ (R_AA2_)	Ascending (descending) and aortic arch resistance	mmHg·cm^−3^·sec
L_AA1_ (L_AA2_)	Ascending (descending) and aortic arch inertance	mmHg·cm^−3^·sec^2^
C_AA1_ (C_AA2_)	Ascending (descending) and aortic arch compliance	mmHg^−1^·cm^−3^
R_THOR_ (R_SupVC_)	Thoracic (superior vena cava) resistance	mmHg·cm^−3^·sec
C_SupVC_	Superior vena cava compliance	mmHg^−1^·cm^−3^
R_AT1_ (R_AB1_)	Thoracic (abdominal) resistance	mmHg·cm^−3^·sec
L_AT1_ (L_AB1_)	Thoracic (abdominal) inertance	mmHg·cm^−3^·sec^2^
C_AT1_ (C_AB1_)	Thoracic (abdominal) compliance	mmHg^−1^·cm^−3^
R_LFA_ (R_LFV_)	Left femoral arterial (venous) resistance	mmHg·cm^−3^·sec
R_RFA_ (R_RFV_)	Right femoral arterial (venous) resistance	mmHg·cm^−3^·sec
C_LLE_ (C_RLE_)	Left (right) femoral compliance	mmHg^−1^·cm^−3^
R_ARM1_ and R_ARM2_ (C_ARM_)	Upper limb resistances (compliance)	mmHg·cm^−3^·sec (mmHg^−1^·cm^−3^)
R_HD1_ and R_HD2_ (C_HD_)	Brain resistances (compliance)	mmHg·cm^−3^·sec (mmHg^−1^·cm^−3^)
R_infVC1_ and R_infVC2_ (C_InfVC_)	Inferior vena cava resistances (compliance)	mmHg·cm^−3^·sec (mmHg^−1^·cm^−3^)
R_HEP1_ and R_HEP2_ (C_HEP_)	Hepatic resistances (compliance)	mmHg·cm^−3^·sec (mmHg^−1^·cm^−3^)
R_KID1_ and R_KID2_ (C_KID_)	Renal resistances (compliance)	mmHg·cm^−3^·sec (mmHg^−1^·cm^−3^)
R_SP1_ and R_SP2_ (C_SP_)	Splanchnic resistances (compliance)	mmHg·cm^−3^·sec (mmHg^−1^·cm^−3^)
R_ABII_ (C_ABII_)	Abdominal (II) resistance (compliance)	mmHg·cm^−3^·sec (mmHg^−1^·cm^−3^)
L_ABII_	Abdominal (II) inertance	mmHg·cm^−3^·sec^2^
R_AbdVC_ (C_AbdVC_)	Abdominal vena cava resistance (compliance)	mmHg·cm^−3^·sec (mmHg^−1^·cm^−3^)
Ees_Left_ (Ees_Right_)	Left (right) ventricular end-systolic elastance	mmHg/ml
Ea_S_ (Ea_P_)	Systemic (pulmonary) arterial elastance	mmHg/ml
Ea_S_/Ees_Left_ (Ees_Right_/Ea_P_)	Left (right) ventricular–arterial coupling	-----

## Data Availability

Not applicable.

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
