# Peer review of "Evaluation of Different Cannulation Strategies for Aortic Arch Surgery Using a Cardiovascular Numerical Simulator"

_bioengineering, 2023, doi:10.3390/bioengineering10010060_

Round 1

Reviewer 1 Report (Previous Reviewer 2)

Comments and Suggestions for Authors
(will be shown to authors)

This simulation study aimed to modify time-varying elastance provides a theoretical background to a three-way cannulation strategy for aortic arch surgery with correlation to the observed clinical practice. This is an interesting study. However, I had some concerns for this study.

Major

1.      Too little data to validate this study should be added as a limitation. How about many individuals used for this study? Please revise accordingly.

2.      Figure 4 showed results of the loop being obtained in basal conditions. I am not familiar with this kind of presentation of results of a loop. Please clarify this concern.

3.      In Table 1, the summary of different items. Please give a rationale about this concern.

4.      The authors should update references.

Author Response

We are grateful to reviewer 1 for his/her comments.

We have acknowledged the limitation of our work in the discussion. We have also included additional references. We feel that 96 references are more than adequate to give an overview about aortic surgery and the use of lumped-parameter modelling.

Figure 4 shows the effect of different cannulation approaches in terms of pressure-volume loops compared to the initial clinical condition of a patient before undergoing cannulation.

Table 1 is quite explicative in a way that we list the symbols and acronyms used with their meaning.

Reviewer 2 Report (Previous Reviewer 3)

Now this paper is ready for publication

Author Response

Thank you very much for your consideration and approval of our manuscript.

Reviewer 3 Report (New Reviewer)

Dear authors:

I appreciate your work very much. The simulation of flow conditions seems to be a valid approach to this complex situation. However, I think it falls short of what is expected in terms of cannulation strategies. In my opinion, you have not selected the preferred clinical cannulation scenarios in this simulation model.

1. in particular, cannulation of the ascending aorta can only serve as a control, as it is the most appropriate perfusion condition for the whole body (RA to ascending aorta).

2. therefore, I would recommend simulating venous outflow from either a) the RA (as a typical site), b) the jugular vein and femoral artery (for "percutaneous" access), c) the femoral artery alone (for re-operation) to the following arteries. to 1. the right axillary artery (to simulate all unilateral scenarios for the brain including the brachiocephalic trunk), 2. both the right axillary artery and the left carotid artery (to simulate bilateral antegrade perfusion). In this particular scenario, simulating the left axillary artery would be of interest because of the left vertebral artery, but this is clinically beside the point. 3. The femoral artery alone (for the emergency scenarios and as a "poor" control for cerebral perfusion due to the long distance). 4. the femoral artery and the right axillary artery (in addition to assumed improved spinal protection, but also to detect a possible concurrent flow situation somewhere in the descending aorta) and possibly 5. the femoral artery with bilateral cerebral perfusion.

3. I am not sure if the model allows the simulation of a clamping of the brachiocephalic trunk. 

4. You can reduce the number of illustrations and focus on the most important ones, because the proposed scenarios are numerous and the reader can hardly cope with such a wealth of information.

5. Can hypothermia be simulated in the model? Perhaps by adjusting the peripheral resistance, the calculated flow at a given temperature and the hydraulic behaviour of the blood, which is a non-Newtonian fluid. If this is possible, then all scenarios could be studied at three or even four temperature levels. I. Normothermia; II. 28°C moderate hypothermia; III. 24° C profound hypothermia. IV. 20° C control 

6. Finally, the figures intended to explain the simulation circuit (2a-c) are rather cluttered and difficult to understand at first sight, even for a doctor familiar with haemodynamics.

Author Response

We are grateful to reviewer 3 for his/her comments.

I appreciate your work very much. The simulation of flow conditions seems to be a valid approach to this complex situation. However, I think it falls short of what is expected in terms of cannulation strategies. In my opinion, you have not selected the preferred clinical cannulation scenarios in this simulation model.

We have addressed the preferred cannulation strategies in the discussion.

  1. in particular, cannulation of the ascending aorta can only serve as a control, as it is the most appropriate perfusion condition for the whole body (RA to ascending aorta).

Yes, we agree with this statement. RA to ascending aorta cannulation remains the standard approach. The aim is the use of a cannulation approach for aortic arch surgery that provides the same benefit of the standard one but with particular focus on the cerebral circulation.

  1. therefore, I would recommend simulating venous outflow from either a) the RA (as a typical site), b) the jugular vein and femoral artery (for "percutaneous" access), c) the femoral artery alone (for re-operation) to the following arteries. to 1. the right axillary artery (to simulate all unilateral scenarios for the brain including the brachiocephalic trunk), 2. both the right axillary artery and the left carotid artery (to simulate bilateral antegrade perfusion). In this particular scenario, simulating the left axillary artery would be of interest because of the left vertebral artery, but this is clinically beside the point. 3. The femoral artery alone (for the emergency scenarios and as a "poor" control for cerebral perfusion due to the long distance). 4. the femoral artery and the right axillary artery (in addition to assumed improved spinal protection, but also to detect a possible concurrent flow situation somewhere in the descending aorta) and possibly 5. the femoral artery with bilateral cerebral perfusion.

The four cannulation configurations address the key issues.

Unilateral or bilateral cerebral perfusion has been an ongoing debate. We believe that bilateral perfusion should always be considered regardless of the completion of the circle of Willis. This is the reason for bilateral carotid cannulation through an end-to-side anastomosis of a 10mm Dacron tubular graft and the motivation for this haemodynamic analysis, which seems to support our clinical practice.

Axillary cannulation is an option in aortic surgery and currently gaining some interest. Nevertheless, it is not free of risks and complications. The cannulation techniques described in this study have been used by Massimo Capoccia (MC), Nick Cheshire (NC) and Ulrich Rosendahl (UR) with successful outcome. It is a sophisticated cannulation approach, which works very well for redo surgery and complex aortic cases requiring prolonged cerebral protection. Here are the three key papers published by MC with his mentors UR and NC who have a long-standing experience with this approach:

Capoccia M, Nienaber CA, Mireskandari M, Sabetai M, Young C, Cheshire NJ, Rosendahl UP. Alternative Approach for Cerebral Protection during Complex Aortic Arch and Redo Surgery. J Cardiovasc Dev Dis 2021; 8(8): 86.

Capoccia M, Pal S, Murphy M, Mireskandari M, Hoschtitzky A, Nienaber CA, Cheshire NJ, Rosendahl UP. Cardiac and Vascular Surgeons for the Treatment of Aortic Disease: A Successful Partnership for Decision-Making and Management of Complex Cases. J Invest Med High Impact Case Rep 2021; 9: 2324709620970890. doi: 10.1177/2324709620970890.

Capoccia M, Mireskandari M, Cheshire NJ, Rosendahl UP. Delayed repair of aortic dissection in sickle cell anaemia as a combined cardiac and vascular surgical approach. J Saudi Heart Assoc 2020; 32(2): 208-212.

The aim was to give a theoretical background to a long-standing clinical approach with a view to further improvement and future studies. The initial simulation results seem to support the use of two key cannulation configurations in our clinical practice. The study is purely aimed at the cannulation approach, which is applicable in the setting of aortic aneurysmal disease and dissection. We are not claiming “magic bullets”. Nevertheless, the two key cannulation configurations discussed in our paper have significantly reduced potential for complications.

  1. I am not sure if the model allows the simulation of a clamping of the brachiocephalic trunk. 

Clamping of the innominate artery is not essential.

  1. You can reduce the number of illustrations and focus on the most important ones, because the proposed scenarios are numerous and the reader can hardly cope with such a wealth of information.

We have already limited the number of illustrations to focus on the key haemodynamic and energetic variables. Further reduction would not give the whole picture.

  1. Can hypothermia be simulated in the model? Perhaps by adjusting the peripheral resistance, the calculated flow at a given temperature and the hydraulic behaviour of the blood, which is a non-Newtonian fluid. If this is possible, then all scenarios could be studied at three or even four temperature levels. I. Normothermia; II. 28°C moderate hypothermia; III. 24° C profound hypothermia. IV. 20° C control 

 The answer is no. Considering that hypothermia is related to peripheral vasoconstriction, adjusting peripheral resistance is an attempt to simulate this condition although not completely reliable. 

  1. Finally, the figures intended to explain the simulation circuit (2a-c) are rather cluttered and difficult to understand at first sight, even for a doctor familiar with haemodynamics.

Figures 2a, 2b and 2c represent a complex cardiovascular network. They are a compromise between excessive simplification and extreme complexity. Further details are available on CARDIOSIM website.

Round 2

Reviewer 1 Report (Previous Reviewer 2)

Thanks for your efforts on revision.

This manuscript is a resubmission of an earlier submission. The following is a list of the peer review reports and author responses from that submission.

Round 1

Reviewer 1 Report

I read the manuscript with interest. I also read the few other publications the authors published in Bioengineeing, J Clin Med, and Computer Methods and Programs in Biomedicine. The approach is presented herein is similar as that of Electrical Lumped Model. This approach although raised some interest few years ago due to its simplicity in providing a rough estimation for cardiac system, it has its own concerns. The authors developed a lumped model for the aorta but unfortunately, I could not find any literature review in the introduction.

1 Introduction needs to better acknowledge the models published so far.

2 This approach models the aorta as a simple elastic model with isotropic small deformation mechanical response. This is far from reality of these tissues.

3The figures are very low quality and sloppy. Please spend more time in creating better quality figures.

4 No validation is provided in this study.

5 How did you define the compliance of the tissues? Try and error?

Author Response

Thank you for your comments.

Although the introduction was aimed to give an overview about aortic disease and its treatment approach as a prelude to the current study, we have given a brief review of the role of lumped parameter models in a clinical setting.

The simulation results are in line with the clinical findings experienced by one of the authors (MC) with particular reference to two cannulation techniques.

The particle/wave behaviour of light and Newtonian mechanics are example of how models can be tailored according to the need and the context in which they are used. Newtonian mechanics allowed moon landing although it does not work in a relativistic field. Therefore, lumped parameter models are quite suited to a clinical setting despite their limitations. The outcome is extremely helpful. The same can be said for pressure-volume analysis, which is quite a powerful approach in cardiovascular physiology.

The original high-definition figures have been uploaded to the journal's website.

Reviewer 2 Report

Major

1.      Too few data to validate of this study should be added as a limitation. How about many individuals used for this study? Please revise accordingly.

2.      Figure 4 showed results of loop is obtained in basal conditions. I am not familiar this kind of presentation of results of loop. Please clarify this concern.

3.      In Table 1, the summary of different item. Please give a rationale about this concern.

Author Response

Thank you for your comments.

Figure 4 shows the effect of different cannulation approaches in terms of pressure-volume loops compared to the initial clinical condition of a patient before undergoing cannulation.

Table 1 is quite explicative in a way that we list the symbols used and their meaning.

We have acknowledged the limitation of our work in the discussion.

Reviewer 3 Report

Dear Authors

I have only minor suggestions.

Why do you decide to not analyze axillary access? I think it is the best way to get an antegrade perfusion.

There are a lot of differences between aneurismal diseases and dissection. Do you analyze these factors?

First cause of stroke during aortic arch repair is embolism during all the maneuvers. Please comment in the discussion.

Do you consider peripheral resistance as a variable factor? 

Author Response

Thank you for your comments.

Axillary cannulation is an option in aortic surgery and currently a sort of fashion. Nevertheless, it is not free of risks. The cannulation techniques described in this study have been used by one of the authors (MC) in clinical practice with successful outcome. It is a sophisticated cannulation approach which works very well for redo surgery and complex aortic cases requiring prolonged cerebral protection. Here are the three key papers published by MC with his mentors Ulrich Rosendahl and Nick Cheshire who have a long-standing experience with this approach:

Capoccia M, Nienaber CA, Mireskandari M, Sabetai M, Young C, Cheshire NJ, Rosendahl UP. Alternative Approach for Cerebral Protection during Complex Aortic Arch and Redo Surgery. J Cardiovasc Dev Dis 2021; 8(8): 86.

Capoccia M, Pal S, Murphy M, Mireskandari M, Hoschtitzky A, Nienaber CA, Cheshire NJ, Rosendahl UP. Cardiac and Vascular Surgeons for the Treatment of Aortic Disease: A Successful Partnership for Decision-Making and Management of Complex Cases. J Invest Med High Impact Case Rep 2021; 9: 2324709620970890. doi: 10.1177/2324709620970890.

Capoccia M, Mireskandari M, Cheshire NJ, Rosendahl UP. Delayed repair of aortic dissection in sickle cell anaemia as a combined cardiac and vascular surgical approach. J Saudi Heart Assoc 2020; 32(2): 208-212.

The aim was to give a theoretical background to a long-standing clinical approach with a view to further improvement and future studies. The initial simulation results confirm the successful practice of two key cannulation techniques. The study is purely aimed at the cannulation approach, which is applicable in the setting of aortic aneurysmal disease and dissection.

Round 2

Reviewer 1 Report

None